# Seasonal Effects of High-Altitude Forest Travel on Cardiovascular Function: An Overlooked Cardiovascular Risk of Forest Activity

**DOI:** 10.3390/ijerph18189472

**Published:** 2021-09-08

**Authors:** Tsung-Ming Tsao, Jing-Shiang Hwang, Ming-Jer Tsai, Sung-Tsun Lin, Charlene Wu, Ta-Chen Su

**Affiliations:** 1The Experimental Forest, National Taiwan University, Nantou 55750, Taiwan; tmtsao@ntu.edu.tw (T.-M.T.); tmj@ntu.edu.tw (M.-J.T.); pine88854@gmail.com (S.-T.L.); 2Institute of Statistical Science, Academia Sinica, Taipei 11529, Taiwan; hwang@sinica.edu.tw; 3School of Forestry and Resource Conservation, National Taiwan University, Taipei 10617, Taiwan; 4Institute of Environmental and Occupational Health Sciences, National Taiwan University College of Public Health, Taipei 10055, Taiwan; 5Global Health Program, National Taiwan University College of Public Health, Taipei 10055, Taiwan; charlenewu@ntu.edu.tw; 6Department of Environmental and Occupational Medicine, National Taiwan University Hospital, Taipei 10002, Taiwan; 7Divisions of Cardiology, Department of Internal Medicine, National Taiwan University Hospital, Taipei 10002, Taiwan

**Keywords:** high altitude, hypoxia, cardiovascular function, acute mountain sickness

## Abstract

Cardiovascular physiological responses involving hypoxemia in low temperature environments at high altitude have yet to be adequately investigated. This study aims to demonstrate the health effects of hypoxemia and temperature changes in cardiovascular functions (CVFs) by comparing intra-individual differences as participants ascend from low (298 m, 21.9 °C) to high altitude (2729 m, 9.5 °C). CVFs were assessed by measuring the arterial pressure waveform according to cuff sphygmomanometer of an oscillometric blood pressure (BP) device. The mean ages of participants in winter and summer were 43.6 and 41.2 years, respectively. The intra-individual brachial systolic, diastolic BP, heart rate, and cardiac output of participants significantly increased, as participants climbed uphill from low to high altitude forest. Following the altitude increase from 298 m to 2729 m, with the atmosphere gradually reducing by 0.24 atm, the measured average SpO_2_ of participants showed a significant reduction from 98.1% to 81.2%. Using mixed effects model, it is evident that in winter, the differences in altitude affects CVFs by significantly increases the systolic BP, heart rate, left ventricular dP/dt max and cardiac output. This study provides evidence that cardiovascular workload increased significantly among acute high-altitude travelers as they ascend from low to high altitude, particularly in winter.

## 1. Introduction

High cardiovascular stress associated with altitude-environment changes is an important emerging public health issue. High-altitude activity involves traveling from a low altitude flat land to a medium-high altitude area of 2500–3500 m became more and more popular in recent decades. Rapid ascension from low to high altitude often causes acute mountain sickness (AMS) [1,2]. The convenience of modern transportation allows for rapid ascent to high altitudes, and can thus compromise acclimatization and expose inexperienced travelers or climbers to the hazards of high altitude [3]. Mountains with an altitude of 8000 feet or higher above sea level are known to increase the risk of mountain sickness; however, this danger is often overlooked. This altitude is also similar to a lot of famous ski resorts, such as the Colorado Rocky Mountain ski resort in the United States and Austrian Alps for skiing or mountaineering in the Western Alps, falling in the middle and high altitudes of 2500–3500 m above sea level. More studies are therefore warranted to demonstrate the health effects of hypoxemia and temperature changes in cardiovascular functions (CVFs) and prevent the occurrence of mountain sickness, including acute cardiopulmonary disease, fatigue, nausea, dizziness, vomiting, and sleep disturbance [4,5,6].

Cardiovascular responses at high altitude are very important for the safety of travelers or climbers. Mountain sickness may be associated with multiple risk factors, such as cold temperature, high-altitude environment (above 2500 m), rate of ascent, hypoxia, altitude reached (especially the sleeping altitude for overnight rest), and individual susceptibility [7,8,9]. In particular, cold temperature and hypoxia are major factors affecting human CVFs because the cardiovascular workload increases with decreasing temperature and hypoxia. An increase of pulmonary artery pressure is a hallmark of the cardiovascular response to high-altitude exposure, which, if pronounced, may lead to cardiovascular morbidity and mortality [10]. The combined mean systolic PAP (right ventricular-to-right atrial pressure gradient) at high altitude (25.3 mmHg) was significantly higher than at low altitude (18.4 mmHg); Arterial oxygen saturation was significantly lower (90.4%) than at low altitude (98.1%). The frequency of thrombotic stroke at high altitude was 93.4% as compared to 79.3% at low altitude. An increased frequency of thrombotic stroke at high altitude was explained by increased hematocrit, which might have caused this in conjunction with hypertension and ischemic heart disease factors [11]. Our previous study discovered increased brachial systolic and diastolic blood pressure (SBP and DBP) and central BP (cSBP and cDBP) in the winter season compared to the summer season [12]. Cooler temperatures induce vasoconstriction to preserve energy for central vital organs, not only leading to increased SBP and DBP, but also central BP and systemic vascular resistance (SVR). The risk of high-altitude illness is associated with increased altitude. Low temperatures at high-altitude were found to be significantly associated with an increased risk of cardiovascular diseases [3,8,13,14,15]. With altitude increases, there are decreases in barometric pressure, which is well-known to be lower at higher altitude. At moderate and high altitude levels, the partial pressure of oxygen is reduced, ultimately leading to alveolar hypoxia. Acute hypoxia causes increases in SBP and DBP, heart rate (HR), cardiac output (CO), catecholamine, impaired HR variability (increased low frequency/high frequency ratio), and SVR [3,4,6,16,17,18,19]. The increase in CO at high altitude is mainly due to an increased HR. Acute hypoxia at high altitude is a potent activator of the sympathetic nervous system. The sympathetic excitation results from acute hypoxia at high altitude and leads to increased HR, BP, and SVR [8,16]. Both hypoxemia and lower temperature at high altitude were found to be significantly associated with an increased risk of AMS and cardiovascular diseases.

However, studies on changes in CVFs and the possible mechanism associated with hypoxemia and temperature changes from low (starting) to high altitude, and rechecking of the CVFs from high to low altitude are still limited. The aims of this study are to demonstrate hypoxemia and temperature variation in CVFs by comparing intra-individual differences from low to high altitude forest in winter and summer, and analyzing the health effects of high vs. low altitude on CVFs and SpO_2_ concentration in winter and summer by using mixed effects model, and discuss the possible mechanisms accounting for the health effects of hypoxemia and temperature variation in cardiovascular hemodynamics at high altitude forest.

## 2. Material and Methods

### 2.1. Design and Participants

The present study was designed to determine the health effects in acute travelers on CVFs from low to high altitude forest in winter and summer. Eleven adult volunteers were recruited in the winter for an observational pilot study, joining a 2-day/1-night high-altitude-environment examination program from 31 December 2016 to 1 January 2017 in National Taiwan University (NTU) Experimental Forest, Nantou County, Taiwan. They gathered in Taipei on 31 December and boarded a medium-sized bus (20 guests) with environmental monitoring and human observation equipment to travel to the wood utilization center in the NTU Experimental Forest, Shuili. Participants stayed at the wood utilization center and provided written informed consent before undergoing cardiovascular health examinations. During the study period of high-altitude health effects from 31 December 2016 to 2 January 2017, they had to maintain dietary control (same diet and environment, with the gender and energy requirements matching the season and altitude). In the morning, participants had to undergo CVFs examinations at the wood utilization center before traveling to the high-altitude mountain (Figure 1).

Participants took a medium-sized bus following the altitude change from Shuili town at altitude 298 m. The travel time was 1.5 h to reach Tataka climber center at altitude 2610 m. Participants arrived at Tataka climber center and took 20 min to rest. Then, we began the first CVFs and SpO_2_ examination (between 9:41 a.m. to 10:18 a.m.) for participants at Tataka climber center. After completing the examination, they traveled 18 min by car to reach Lulin villa at altitude 2729 m. Then, the participants received the first CVFs and SpO_2_ examination at 11:06 a.m. on arrival at Lulin villa. They rested at the garden platform of Lulin villa and then received a second CVFs and SpO_2_ examination after resting for 60 min. After that, participants came back to the starting point at Tataka climber center and underwent the second CVFs and SpO_2_ examinations at 12:46 p.m., with about a 2 h interval between the first to second examinations. After completing the high-altitude study, participants took the same bus and travelled downhill to Wuqi district at altitude 10 m where they again received CVFs and SpO_2_ examination at 16:03 p.m. (Figure 2).

In summer, sixteen adult volunteers were recruited for joining one day high-altitude health effects study on 23 June 2017. The subjects arrived at the wood utilization center at 9:45 a.m., and received the first CVFs and SpO_2_ measurements at 10:23 a.m. After completing the examination, the participants traveled by car to reach Tataka climber center. After taking a rest for 20 min, they received the first CVFs and SpO_2_ measurements at 15:14 p.m. The participants then traveled by car to Lulin villa (approximate travel time is 18 min), and received the CVFs and SpO_2_ examination at 16:05 p.m. Afterwards, participants came back to the Tataka climber center and underwent the second CVFs and SpO_2_ measurements at 17:02 p.m. Finally, the study subjects took the same bus and travelled downhill to Heshe (alt. 760 m) where they again received CVFs and SpO_2_ examination at 18:43 p.m. (Figure 2).

### 2.2. Environmental Assessment

The instruments used for environmental monitoring were real-time recording at low and high-altitude environments in winter and summer. The concentration and size distribution, as well as the real-time mass concentration of particulate matter (PM_10_, PM_2.5_, PM_1.0_), were monitored using a DustTrak aerosol monitor (model 8533; TSI Inc., Shoreview, MN, USA). The temperature and relative humidity were monitored using an IAQ monitor (model 2211; Kanomax, Andover, NJ, USA). Oxygen saturation rate (finger SpO_2_ value) was obtained using the Rossmax Pulse Oximeter, Model SB100 (Rossmax International Ltd., Taipei, Taiwan).

### 2.3. Cardiac and Vascular Functions Assessments

An oscillometric BP device (DynaPulse 200M, Pulse Metric Inc., San Diego, CA, USA) was used to record the arterial pressure waveform using a cuff sphygmomanometer [20,21]. BP was measured twice (left and right hands) after at least 5 min of rest in a sitting position in a quiet room. BP was determined by changes in pressure waveform according to Bernoulli flow effects. Central SBP and DBP, vascular compliance and peripheral resistance of the brachial artery were derived by incorporating the arterial pressure signals from a standard cuff sphygmomanometer using a physical model. The BP used in the analyses was compared intra-individually in left and right arm for each subject, thus two measurements for each participant. This method has been employed to derive other cardiac hemodynamic parameters, such as the maximum rate of left ventricular pressure increase (LV dP/dt max), stroke volume (SV), CO, and cardiac index (CI); and its application was also validated in our recent studies [12,22,23]. The data were electronically transmitted from the collection site to a central analysis center.

### 2.4. Statistical Analysis

We first examined the data by testing any statistical differences between two groups. Student’s two-tailed *t*-test or the Mann–Whitney U-test if not in normal distribution was used to compare the means ± standard deviations (SD) of continuous variables between different altitudes. The paired *t*-test was used to evaluate each individual’s mean difference in cardiac and vascular hemodynamics measured at two low altitude and two high altitude environments. CVFs changes in different environments from low altitude to high altitude were tested using the paired *t*-test. All statistical analyses were performed with SAS statistical software (version 9.4, SAS Institute Inc., Cary, NC, USA).

We then fit a mixed effects model to examine the effects of high altitude on CVFs changes in winter and summer. Let $Y_{ijk}$ be the response variable of the model denoting the *i*-th participant’s CVFs measured in season *j* (summer or winter) at site *k* (Shuili, Tataka, Lulin). The explanatory variables of interest included three dummy variables: (1) W = 1 if measured in winter, (2) HS = 1 if measured at the high-altitude sites of Tataka and Lulin (alt. >2500 m) in summer, (3) HW = 1 if measured at the two high-altitude sites in winter. Specifically, the model is written as
(1)$$Y_{ijk}=\alpha +a_{i}+\beta_{1}\times W+\beta_{2}\times HW+\beta_{3}\times HS+\underset{l}{\sum }\left(\gamma_{l}\times X_{il}\right)+\delta \times Z_{jk}+\varepsilon_{ijk},$$
where the characteristic variables $X_{il}$ for the i-th participant include age, sex, BMI, any disease of HTN and DM, HLP, drinking and regular exercise, and Z*_jk_* is relative humidity. The random component $a_{i}\sim N\left(0,\;\sigma_{a}^{2}\right)$ is to model variation among the participants caused by unmeasured factors. Finally, for taking account of repeated measurements of each participant, we also assumed the measurements of each participant were correlated in the model. That is, we assumed that $\varepsilon_{ijk}\sim N\left(0,\;\sigma_{a}^{2}\right)$ and $cor(\varepsilon_{ijk_{1}},\;\varepsilon_{ijk_{2}})=\rho$ when $k_{1}\neq k_{2}$. The coefficient $\beta_{1}$ is for adjusting the CVFs difference at the baseline site of Shuili (alt. 298 m) between winter and summer. The coefficients $\beta_{2}$ and $\beta_{3}$ represent CVFs changes from low altitude to high altitude in summer and winter, respectively. The R package was used to obtain the model estimates and standard errors for each response variable of CVFs.

## 3. Results

The general characteristics of the participants are summarized in Table 1. The mean ages of participants in winter and summer were 43.6 and 41.2 years, respectively. The participants in summer consisted of more males (87.5%) compared to wintertime (54.6%). The prevalence of hypertension (≥140 mmHg in systolic BP or ≥90 mmHg in diastolic BP) in winter and summer was 36.4% and 6.3%, respectively. Hypertensive adults taking medication accounted for 18.2% of the participants, and 6.3% were diagnosed with hypercholesterolemia.

The environmental monitoring results in winter and summer (PM_1_, PM_2.5_, PM_10_, temperature, relative humidity, and atmospheric pressure) in low altitude city and high altitude mountain are presented (Table 2). PM concentrations at high altitude in winter were lower than that at low altitude. PM concentrations were relatively high at the wood utilization center (wood factory) and Wuqi, which is close to the seaside industrial area. Moreover, during summer, PM concentration was relatively lower than that in winter. The starting temperature at low altitude in Shuili town was 21.9 °C. When participants reached high altitude Tataka and Lulin villa, the final temperature was reduced to 11.4 °C and 9.5 °C, respectively (Table 2). Temperature in Lulin villa at high altitude in summer was 16.5 ± 0.66 °C which was higher than that in winter (9.5 °C).

Table 3 presents comparisons of cardiac and vascular function using the DynaPulse monitoring device in different altitude towns and mountains. Participants took a medium-sized bus (20 guests) from low to high altitude forest. Following the altitude change from Sui-Li town (alt. 298 m) to Tataka (alt. 2610 m) and the final stop at Lulin villa (alt. 2729 m), the corresponding change in BP was in winter as follows: the intra-individual comparisons showed brachial SBP and DBP, and cSBP significantly increased from 120.4, 78.1, and 128.8 mmHg at Sui-Li to 136.1, 82.6, and 142.9 mmHg at Tataka, and finally 141.1, 85.8, and 148.2 mmHg respectively at Lulin villa. In contrast to the winter season, the BP components and CVFs was not significantly increased at Shuili compared to those at Tataka and Lulin villa in summer season.

The same trend was found in cardiac function for participants traveling from Shuili to Tataka and Lulin villa. Cardiac function including HR, LV dP/dt max, LV contractility, CO, and CI at Shuili was lower than at Tataka and Lulin villa, and the differences were significant. Following the altitude change from Shuili town to Tataka and Lulin villa, the corresponding changes in cardiac function in winter were as follows: the intra-individual comparisons showed HR and CO, and LV dP/dt max significantly increased from 69.2 beats/min, 4.6 L/min, and 1116.0 mmHg/s at Sui-Li to 83.8 beats/min, 5.9 L/min, and 1419.0 mmHg/s at Tataka, and finally 88.4 beats/min, 6.0 L/min, and 1463.0 mmHg/s, respectively, at Lulin villa. In summer, HR and CO, and LV dP/dt max significantly increased from 71.5 beats/min, 5.2 L/min, and 1182.0 mmHg/s at Suili to 81.5 beats/min, 6.4 L/min, and 1359.0 mmHg/s at Tataka, and finally 81.4 beats/min, 6.0 L/min, and 1341.0 mmHg/s, respectively, at Lulin villa. Furthermore, vascular function revealed significantly reduced vascular compliance. The SVC and brachial artery distensibility of participants at Shuili in winter were 1.2 mL/mmHg and 6.1%/mmHg respectively, which were higher than in Tataka (1.1 mL/mmHg and 5.3%/mmHg), and Lulin villa (1.0 mL/mmHg and 5.1%/mmHg), respectively.

Following the altitude increase (alt. 2431 m) and the gradually reduced atmosphere (0.24 atm), the average measured SpO_2_ of participants in winter showed a significant reduction (16.9%). The measurement in Lulin villa was significantly lower (81.2 ± 8.0%) than those of participants in Shuili (98.1 ± 0.9%). In summer, the average measured SpO_2_ of participants showed only reduction (7.9%). The measurement in Lulin villa was lower (89.8 ± 3.0%) than those of participants in Shuili (97.7 ± 0.7%). The average measured SpO_2_ of participants in Lulin villa in winter was lower (81.2 ± 8.0%) than those of participants in summer (89.8 ± 3.0%). CVFs showed no significant differences between Tataka and Lulin villa because of the minor altitude differences (147 m) between these two locations.

Table 4 showed, after staying 1 h in Lulin villa and going back to low altitude, the brachial SBP and DBP, and cSBP of participants in Wuqi district in winter were significantly lower than those of participants at Lulin villa. The MAP, PP, and HR of participants in winter were significantly lower at Wuqi district than at Lulin villa. In summer, the brachial SBP, and cSBP of participants at Heshe were significantly lower than those of participants at Lulin villa. The HR of participants in summer were significantly lower at Heshe than at Lulin villa. Following the altitude change from Lulin villa to Shuili town, the corresponding changes in cardiac function in winter were as follows: the intra-individual comparisons showed the HR and CO, and LV dP/dt max significantly decreased from 87.5 beats/min, 5.9 L/min, and 1383.1 mmHg/s at Lu-Lin villa to 73.8 beats/min, 5.0 L/min, and 1196.8 mmHg/s, respectively at Wuqi district. In summer, the HR and CO, and LV dP/dt max significantly decreased from 80.2 beats/min, 6.2 L/min, and 1340.0 mmHg/s at Lu-Lin villa to 69.7 beats/min, 5.1 L/min, and 1158.0 mmHg/s, respectively at Heshe. When participants returned to the low altitude Wuqi district, the SpO_2_ of participants in winter became higher (97.7 ± 1.4%) than when in Lulin villa (83.8 ± 5.1%), displaying statistically significant differences between both. In summer, the average measured SpO_2_ of participants at Heshe became significantly higher (96.8 ± 2.1%) than when in Lulin villa (89.9 ± 3.5%).

Table 5 presented an adaptation of the cardiovascular hemodynamics in the high-altitude mountain. After staying 2 h at Tataka, BP components such as brachial SBP, DBP, and cSBP among of participants were significantly lower, indicating a rapid physiological adaptation of human body exposure at high altitude. However, there was no significant effect on cardiovascular adaptation at Lulin villa after staying only for 1 h. Vascular functions after staying 2 h at Tataka were as follows: the intra-individual comparisons showed CO, and LV dP/dt max decreased significantly from 5.9 L/min, and 1419.1 mmHg/s to 5.7 L/min, and 1242.4 mmHg/s, respectively. The SVC and brachial artery distensibility of participants at Tataka district were higher, revealing statistically significant differences.

In Table 6, the mixed effects model showed the health effects of high vs. low altitude on CVFs and SpO_2_ concentration in winter and summer. After controlling for seasons, altitudes, age, gender, BMI, hypertension, habits of smoking and alcohol, and hypercholesterolemia, the effects of high altitude vs. low altitude on CVFs in winter are as follows: increased 9.2, and 3.3 mmHg in the SBP and DBP, respectively, increased 7.6, and 2.7 mmHg in the cSBP and cDBP, respectively; and increased 16.5 beats/min, and 1.4 L/m in the HR and CO, respectively. The SpO_2_ concentration significantly decreased by −8.3 ± 1.47% at high and low altitude in winter. There were no statistical significant differences between high altitude and low altitude in summer, with the exception of HR and CO, which showed significant increase (9.5 ± 4.24% beats/min and 1.3 ± 0.46% L/min, respectively); the SpO_2_ concentration significantly decrease by −7.9 ± 2.84%.

## 4. Discussion

This study provides first line evidence of significantly increased cardiovascular workload from low to high altitude forest in acute high altitude travelers. On the contrary, we also demonstrated reduced cardiovascular workloads among travelers went downhill from high altitude to low altitude forest. Hypoxemia in high altitude mountains is credited as the major physiological basis of the cardiovascular response in humans. AMS, which may develop at altitudes over 2500 m, is a potentially life threatening illness, and can lead to high altitude cerebral edema and pulmonary edema [8]. The prevention of AMS for many people engaging in mountain tourism in famous mountain areas of the world, particularly those of a one-day or two-day trip can have far-reaching public health implications.

The novel finding of this study was the BP components (including central BP); and cardiac workload parameters (HR, SV, LV contractility, CO, and CI) were significantly higher in Tataka and Lulin villa. The SpO_2_ of participants in Lulin villa at an altitude 2729 m was significantly lower (81.2 ± 8.0%) than those of participants in Shuili town (98.1 ± 0.9%). Mountain sickness may be associated with multiple risk factors, such as lower temperature, high altitude above 2500 m, rate of ascent, hypoxia, altitude reached (especially the sleeping altitude), and individual susceptibility [7,8]. Prior study showed as a group of 139 healthy young males from 500 m rapid ascent by train, the HR, ejection fraction, fractional shortening, SV, and CO were significantly increased within 24 h of arrival at 3700 m [3]. The HR, SBP, DBP, and mean artery pressure were significantly increased on the seventh day of acclimatization at 4400 m high altitude to compare with 3700 m of baseline level in all subjects. However, the SV, SVI, CO, and CI were significantly decreased. Lower temperature and hypoxia directly affect multiple CVFs. The BP and HR of 27 healthy normotensive subjects (8 children, 9 adults, and 10 elderly subjects) of a wide age range (6–83 years) significant increased exposure from lower altitude (200 m) to high altitude (2950 m) rapid ascent by cable car in 30 min [19]. It is evident that whether it is a healthy subject or a patient with hypertension and hyperlipidemia in this study, rapid ascension from low to high altitude same causes the risk of cardiovascular load. Our study findings are compatible with previous studies showing acute climbing or driving into cooler and high mountain environments can result in significantly increased BP and HR [4,6,16,17].

The cSBP and cDBP information is very important for understanding coronary perfusion during diastole of the aortic pressure and systolic BP during systole of left ventricle of heart. It could be used as a reliable predictor of cardiovascular morbidity and mortality [24]. Many studies demonstrated cooler temperatures in winter increase arterial BP among hypertensive and healthy subjects such as the elderly, adults, and children [25,26]. Tsao et al. [12] demonstrated a significant decrease in brachial and central BP among 72 staff members. Results showed an average decrease in the SBP of 0.9 ± 0.19 mmHg and DBP of 0.8 ± 0.15 mmHg per 1 °C increase, in the seasonal variation of temperature change after controlling covariates; similar changes were observed in cSBP and cDBP of 0.8 ± 0.15 mmHg. Low temperature is the most important environmental factor for cardiovascular load. Its influence on the course and onset of cardiovascular events in winter are not well understood [27]. The proposed mechanisms included the effects of thermoregulatory responses on vasoconstriction, blood flow resistance, and high BP. It is also possible to increase the risk of thrombosis through an increased RBC count, blood viscosity, and platelet count while subjects are exposed to cold temperatures.

Hypoxia is one of the main causes of mountain sickness, affecting people who climb or live in different mountainous areas of the world. A 1% increase in oxygen concentration is the equivalent of descending by 300 m in altitude [28]. Rao et al. [3] reported the S_P_O_2_ value was significantly reduced in all subjects (139 healthy young males) from 98.4% to 88.9% at resting state at 500 m upon acute high-altitude at 4400 m exposure. A total of 75 subjects met the criteria for AMS within 24 h after arrival at 3700 m. In this study, it is very important to highlight the dramatic reduction in blood oxygen concentration (SpO_2_ dropped from 98.1% to 81.2%) and significant increase in HR significantly by 19.2 beats/min (from 69.2/min to 88.4/min, about increased by 27.6%) when participants arrived at Lulin villa (Figure 3). CO also increased by nearly 30.4% accordingly. High altitude environment provides a field experiment to study the association between hypoxemia and cardiovascular risk. Hypoxia at high altitude increases risk of cardiovascular disease. It has been hypothesized that hypoxemia is responsible for detrimental cardiovascular outcomes [15]. Many reports showed that hypoxemia can lead to stroke, heart failure, and cardiovascular decompensation in the Peruvian Andes [29,30,31]. De Ferrari et al. [32] reported that excessive erythrocytosis was associated with higher odds of having cardiometabolic syndrome, it is the hallmark of chronic mountain sickness. Studies have demonstrated that intermittent hypoxemia leads to low-grade systemic inflammation and insulin resistance [33,34]. High altitude-related hypoxia is associated with physiological changes in the cardiovascular system including activation of the sympathetic system [35]. Most of these changes increase cardiac work, which in turn may increase cardiovascular disease risk. The effect of hypoxia on cardiovascular disease response to high altitude, both acute and chronic, has been reviewed recently in detail [4,36]. Hypoxia may cause alterations in gut microbiota, which may cause changes in gut immunity, particularly Peyer’s patches, NK cells and inflammatory cytokines, secretary immunoglobulins, and gut microbiota [37].

Our study results have profound implications for public health, as we demonstrated that to protect travelers from the cold winter in high altitude mountains, adjustments in temperature and eating habits, especially diet control, is important [38]. In this study, for preventing AMS, each subject had to receive same dietary control for breakfast and lunch, including providing higher protein (beef, shrimp, beans, and milk), eating higher carbohydrates (cereals, potato foods), foods that help digestion (Miso soup, tangerine peel, cheese etc.), and more drinking water. Energy consumption and thermal energy requirements are higher at high altitude than at low altitudes [39]. Travelers’ protein catabolism will increase at high altitude, if the amount of protein synthesis is less than the amount of degradation, chronic digestive diseases will occur, which may easily cause tissue trauma and starvation. When digestive function declines, pepsin activity decreases, which results in loss of appetite, nausea, vomiting, bloating, abdominal pain, and diarrhea. Travelers must increase their intake of animal protein like milk, yolk, beans, sesame foods, and kumquat or bergamot to prevent flatulence. In addition, at high altitude, blood sugar decreases, sugar catabolism and heme both increase, and travelers must increase intake of high carbohydrate, vitamin, and iron-rich food. For proactive prevention of AMS, travelers must take a rest day every three or four days when trekking above 3000 m to minimize the risk of AMS and allow acclimatization. The development of hypoxemia and higher cardiovascular workloads in acute mountain travelers requires further epidemiologic research.

The present findings offer some possible explanations for the physiological needs of the human body while being exposed to a high-altitude hypoxia environment. First, this study confirmed for low altitude to high altitude mountaineering, the human body is faced with cold weather, low atmosphere, and significant hypoxia. Therefore, the heart adapts by increasing BP, HR, and CO, as well as increasing the cardiovascular workload to combat the risk of hypoxia (Figure 4), if one makes a short stop (even for only two hours) during the climbing process, the excessive workload of the cardiovascular system can be attenuated. Thirdly, there are many high mountains in Taiwan that exceed 8000 feet above sea level. This height is the critical point of being more prone to high mountain disease. It turns out, above this level, the blood oxygen concentration of the human body will fall to 90% or less. At sea level, if the patient’s blood oxygen in the hospital falls below 90%, it is life-threatening. Just like in this study, it is very easy to reach the state of hypoxemia with acute high attitude exposure in the cold winter season. This study points to the potential risk of “mountain day trips” in Taiwan and other countries, which are quite noteworthy, particularly among the susceptible population, such as those with cardiovascular diseases, airway obstruction, asthma, uncontrolled high BP, chronic disease, and the elderly.

The strength of this study can be confirmed by the consistent findings on different aspects of the cardiovascular system, including BP components, cardiac function, and vascular workloads. The cardiovascular response in uphill climbing being reconfirmed by going downhill also corroborated our findings and the physiologic basis of hypoxia at high altitude, subsequently increasing the cardiac and vascular workloads in acute high altitude climbers. The findings of two hours of rest at high altitude significantly attenuating the cardiovascular overload also demonstrated the adaptation potential of cardiovascular systems, providing evidence of the necessity of gradually ascending high mountains when carrying out climbing activities, particularly at mountains higher than 8000 feet above sea level.

Limitations should also be disclosed. First, we did not consider dietary habits during high mountain travel. Secondly, this study had a small sample size. Thirdly, this study did not measure blood oxygenation, with SpO_2_ only providing an indirect value of hypoxemia.

## 5. Conclusions

This study demonstrated hypoxemia and cold temperature significantly increased cardiovascular workloads in acute high-altitude travelers. The significant increase in cardiovascular loads was accompanied by a marked decrease in SpO_2_ and low temperature at high altitude. The cardiovascular adaptation after a 2-h short stay in a high-altitude environment is important to attenuate the rapid increase in CO in a hypoxia environment. For proactive prevention of AMS, future investigations could focus on the association between developing hypoxemia and higher cardiovascular workloads in acute mountain travelers.

## Figures and Tables

**Figure 1 ijerph-18-09472-f001:**
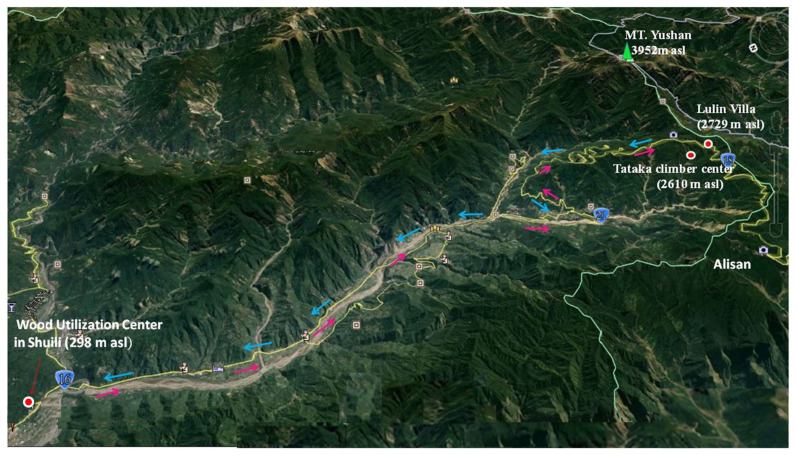
Route map of forest travel from low to high altitude.

**Figure 2 ijerph-18-09472-f002:**
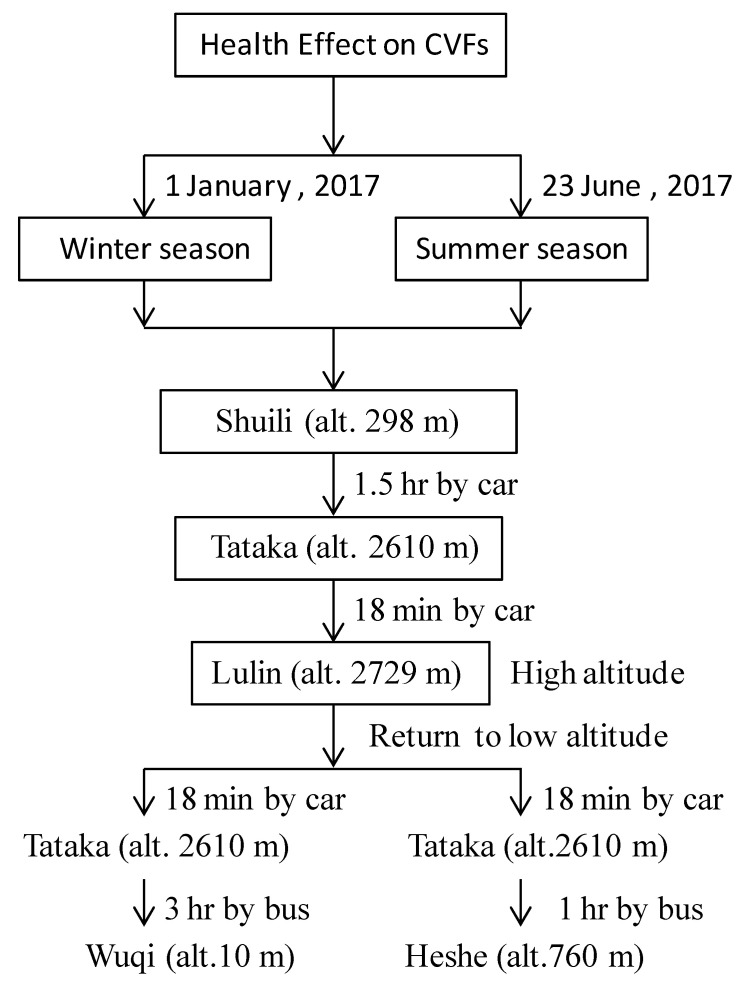
Flow chart illustrating the time points and altitudes of examination from low (starting) to high altitude, and rechecking of CVFs from high to low altitude.

**Figure 3 ijerph-18-09472-f003:**
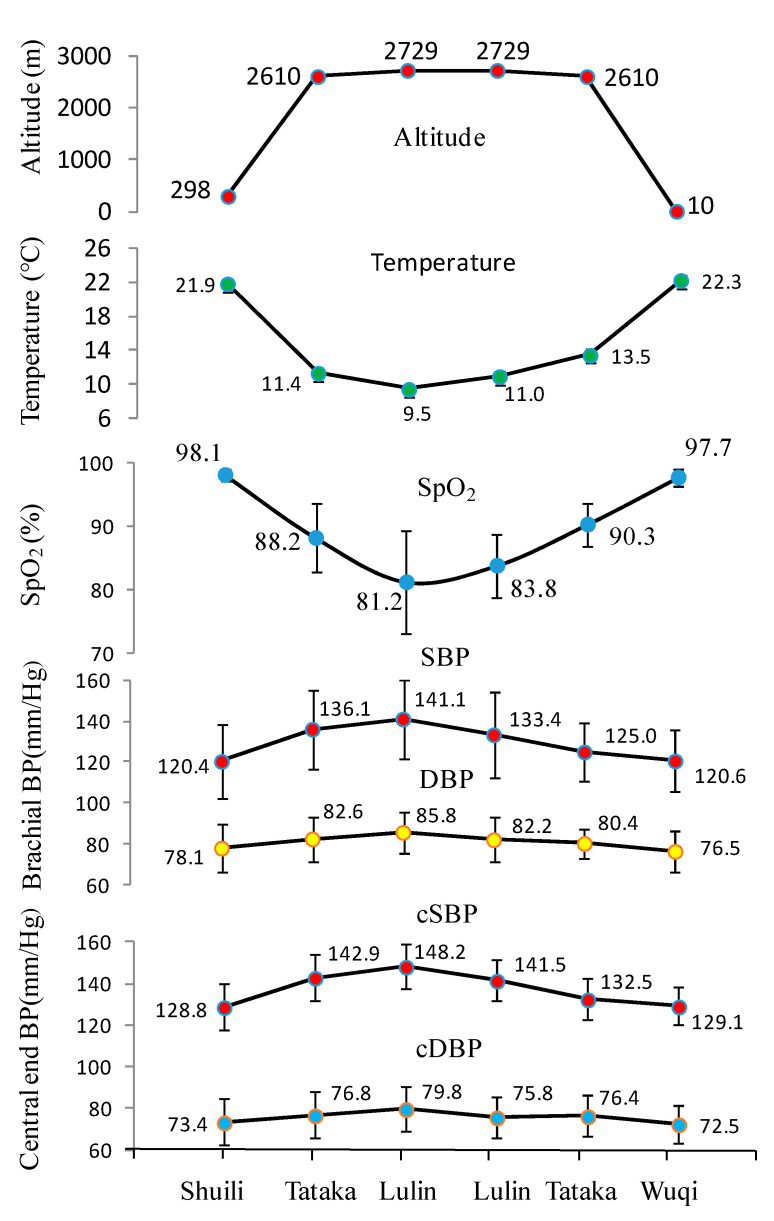
Variational comparison of the oxygen saturation rate and cardiac function after short-term exposure to high altitudes upon return to low altitudes.

**Figure 4 ijerph-18-09472-f004:**
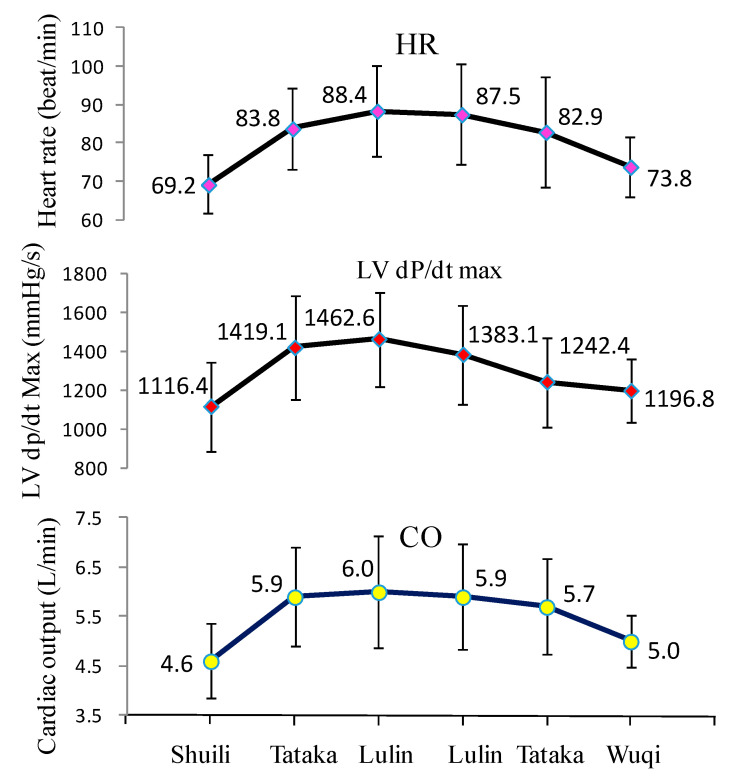
Variational comparison of the oxygen saturation rate and cardiac function after short-term exposure to high altitudes upon return to low altitudes.

**Table 1 ijerph-18-09472-t001:** General characteristics among participants

	Winter Winter	Summer Summer
Variables	January 2017	June 2017
	(*n* = 11)	(*n* = 16)
Age (years)	43.6 ± 11.6	41.2 ±3.6
Male sex (%)	54.6	87.5
BMI (kg/m^2^)	25.0 ± 4.0	25.8 ± 1.1
Smoking habit (%)	0	12.5
Alcohol drinking habit (%)	18.2	37.5
Diabetes (%)	9.1	0
Hypertension (%)	36.4	6.3
Hypercholesterolemia with medication (%)	18.2	6.3

**Table 2 ijerph-18-09472-t002:** Environmental and air quality monitoring.

**Winter**		**Shuili** ***N* = 107**	**Tataka before** ***N* = 54**	**Lulin** ***N* = 40**	**Tataka after** ***N* = 23**	**Wuqi** ***N* = 102**
PM_1_	μg/m^3^	58.16 ± 3.10	11.56 ± 2.75	0.13 ± 0.33	6.17 ± 2.52	76.71 ± 20.63
PM_2.5_	μg/m^3^	59.10 ± 3.18	12.38 ± 2.71	0.13 ± 0.33	6.35 ± 2.53	79.50 ± 21.48
PM_10_	μg/m^3^	64.88 ± 4.23	21.13 ± 4.22	0.23 ± 0.48	10.57 ± 3.72	88.07 ± 24.42
Temperature	°C	21.90 ± 0.20	11.43 ± 0.48	9.48 ± 0.41	13.50 ± 0.63	22.29 ± 0.50
Relative H.	%	71.49 ± 1.52	63.48 ± 2.96	68.87 ± 1.74	64.39 ± 1.21	82.86 ± 2.43
Atmosphere	atm	0.97	0.74	0.73	0.74	1.0
**Summer**		**Shuili** ***N* = 126**	**Tataka before** ***N* = 39**	**Lulin** ***N* = 41**	**Tataka after** ***N* = 17**	**Heshe** ***N* = 65**
PM_1_	μg/m^3^	21.67 ± 1.81	17.15 ± 7.02	15.50 ± 5.56	18.12 ± 0.70	15.42 ± 1.26
PM_2.5_	μg/m^3^	22.14 ± 1.94	18.00 ± 7.05	15.85 ± 5.96	18.18 ± 0.73	15.63 ± 1.16
PM_10_	μg/m^3^	24.52 ± 2.42	20.92 ± 11.93	19.74 ± 14.99	19.00 ± 0.71	17.00 ± 1.41
Temperature	°C	30.69 ± 0.13	17.06 ± 1.69	16.53 ± 0.66	15.30 ± 0.31	25.3 ± 0.89
Relative H.	%	67.62 ± 0.75	87.74 ± 9.18	83.64 ± 3.06	91.37 ± 1.89	78.3 ± 1.03
Atmosphere	atm	0.97	0.74	0.73	0.74	1.0

**Table 3 ijerph-18-09472-t003:** Comparisons of cardiac function and hemodynamic parameters ascending from low to high altitude forest in winter and summer.

	Winter (N = 22)					Summer (N = 32)				
	1. Shuili (298 m)	2. Tataka (2610 m)	3. Lulin (2729 m)	P_1_ 1 vs. 2	P_2_ 1 vs. 3	4. Shuili (298 m)	5. Tataka (2610 m)	6. Lulin (2729 m)	P_3_ 4 vs. 5	P_4_ 4 vs. 6
**BP Components**										
SBP (mmHg)	120.4 ± 17.6	136.1 ± 19.3	141.1 ± 19.3	<0.0001	<0.0001	120.7 ± 13.2	123.6 ± 17.0	129.3 ± 16.6	0.0786	<0.0001
DBP (mmHg)	78.1 ± 11.6	82.6 ± 10.9	85.8 ± 9.9	0.0096	0.0034	78.9 ± 8.9	76.2 ± 8.9	82.2 ± 10.7	0.0022	0.0011
cSBP (mmHg)	128.8 ± 17.5	142.9 ± 18.2	148.2 ± 18.4	<0.0001	<0.0001	129.1 ± 15.5	131.3 ± 18.0	137.3 ± 16.4	0.2298	<0.0001
cDBP (mmHg)	73.4 ± 11.2	76.8 ± 11.0	79.8 ± 10.4	0.0781	0.0530	75.0 ± 9.1	72.0 ± 9.4	78.0 ± 10.1	0.0052	<0.0001
MAP (mmHg)	90.9 ± 12.3	97.7 ± 12.4	101.3 ± 12.2	0.0001	0.0005	92.4 ± 10.3	91.8 ± 11.9	97.5 ± 11.8	0.5742	0.0042
PP (mmHg)	55.3 ± 11.3	66.1 ± 11.3	68.4 ± 9.9	0.0026	0.0015	54.1 ± 11.0	59.3 ± 12.3	59.4 ± 9.7	0.0065	<0.0001
HR (beats/min)	69.2 ± 7.6	83.8 ± 10.7	88.4 ± 11.8	<0.0001	<0.0001	71.5 ± 9.1	81.5 ± 10.7	81.4 ± 10.0	<0.0001	0.008
**Cardiac Function**										
LVE (sec)	0.29 ± 0.04	0.24 ± 0.02	0.22 ± 0.03	<0.0001	<0.0001	0.3 ± 0.004	0.26 ± 0.04	0.24 ± 0.02	<0.0001	<0.0001
LV Max (mmHg/s)	1116 ± 230	1419 ± 263	1463 ± 240	<0.0001	<0.0001	1182 ± 213	1359 ± 281	1341 ± 199	<0.0001	<0.0001
LVC (1/s)	14.0 ± 1.9	16.4 ± 1.9	16.3 ± 1.6	0.0001	0.0003	15.0 ± 1.8	17.0 ± 1.9	16.1 ± 1.4	<0.0001	<0.0001
CO (L/min)	4.6 ± 0.75	5.9 ± 1.02	6.0 ± 1.1	<0.0001	<0.0001	5.2 ± 1.0	6.4 ± 1.8	6.0 ± 1.4	<0.0001	<0.0001
CI (L/min/m^2^)	2.6 ± 0.4	3.45 ± 0.6	3.6 ± 0.7	<0.0001	<0.0001	2.8 ± 0.4	3.4 ± 0.7	3.2 ± 0.6	<0.0001	<0.0001
SV (mL)	65.4 ± 11.0	69.4 ± 9.8	67.5 ± 6.8	0.0173	0.0488	70.4 ± 12.4	76.0 ± 15.1	72.7 ± 13.8	<0.0001	0.0074
SVI (mL/m^2^)	36.8 ± 4.9	40.3 ± 5.1	39.7 ± 4.7	0.0178	0.0544	38.0 ± 3.9	41.1 ± 4.7	39.3 ± 4.6	<0.0001	0.0093
Vascular Function										
SVC (mL/mmHg)	1.2 ± 0.24	1.07 ± 0.18	1.0 ± 0.1	0.0146	0.0017	1.3 ± 0.3	1.3 ± 0.3	1.3 ± 0.3	0.6611	0.0209
SVR (dynes/sec/cm^5^)	1654 ± 354	1351 ± 245	1376 ± 263	<0.0001	<0.0001	1486 ± 319	1213 ± 306	1363 ± 327	<0.0001	0.0016
BAC (mL/mmHg)	0.07 ± 0.03	0.06 ± 0.02	0.07 ± 0.02	0.1733	0.0349	0.08 ± 0.02	0.08 ± 0.02	0.08 ± 0.02	0.2167	0.5220
BAD (%/mmHg)	6.1 ± 1.79	5.3 ± 1.0	5.1 ± 0.8	0.0586	0.0072	6.1 ± 1.3	6.0 ± 1.4	5.8 ± 1.0	0.4023	0.0935
BAR (dynes/sec/cm^5^)	212 ± 107	207 ± 104	175 ± 80	0.0447	0.0488	158 ± 57	152 ± 72	146 ± 64	0.3395	0.0475
**SpO_2_ (%)**	98.1 ± 0.9	88.17 ± 5.46	81.2 ± 8.0	<0.001	<0.001	97.7 ± 0.7	90.8 ± 2.8	89.8 ± 3.0	<0.0001	0.0002

SBP: systolic blood pressure; DBP: diastolic blood pressure; cSBP: central end-SBP; cDBP: central end-DBP; MAP: mean artery pressure; PP: pulse pressure; HR: heart rate; LVE:LV ejection time; LV Max: Left ventricular (dp/dt Max); LVC: Left ventricular contractility; CO: cardiac output; CI: cardiac index; SV: stroke volume; SVI: stroke volume index; SVC: systemic vascular compliance; SVR: systemic vascular resistance; BAC: brachial artery compliance; BAD: brachial artery distensibility; BAR: Brachial artery resistance.

**Table 4 ijerph-18-09472-t004:** Comparisons of cardiac function and hemodynamic parameters descending from high to low altitude in winter and summer.

	Winter (N = 22)					Summer (N = 32)				
	1. Lulin after (2729 m)	2. Tataka after (2610 m)	3.Wuqi (10 m)	P_1_ 1 vs. 3	P_2_ 2 vs. 3	4. Lulin after (2729 m)	5. Tataka after (2610 m)	6. Heshe (760 m)	P_3_ 4 vs. 6	P_4_ 5 vs. 6
**BP Components**										
SBP (mmHg)	133.4 ± 20.8	125.0 ± 14.4	120.6 ± 15.0	<0.0001	0.0774	131 ± 18	127.9 ± 17.8	124.4 ± 18.1	0.0150	0.2663
DBP (mmHg)	82.2 ± 10.7	80.4 ± 7.2	76.5 ± 10.4	0.0016	0.0176	82.4 ± 12.2	83.6 ± 8.9	82.6 ± 12.4	0.9119	0.6388
cSBP (mmHg)	141.5 ± 21.5	132.5 ± 16.4	129.1 ± 14.8	<0.0001	0.1840	138 ± 18	135.0 ± 17.4	130.7 ± 18.5	0.0050	0.2160
cDBP(mmHg)	75.8 ± 10.1	76.4 ± 9.8	72.5 ± 8.9	0.0692	0.0325	78.1 ± 11.3	79.3 ± 9.0	77.5 ± 11.80	0.6859	0.4138
MAP (mmHg)	96.1 ± 12.8	94.4 ± 10.2	90.8 ± 10.3	0.0072	0.0450	97.3 ± 12.8	96.6 ± 10.7	94.3 ± 13.06	0.0329	0.3308
PP (mmHg)	65.7 ± 14.2	56.1 ± 12.2	56.6 ± 9.4	0.0008	0.8062	59.9 ± 10.3	55.7 ± 11.1	53.2 ± 11.08	0.0141	0.3001
HR (beats/min)	87.5 ± 13.0	82.9 ± 14.2	73.8 ± 8.0	<0.0001	<0.0001	80.2 ± 4.8	74.2 ± 7.9	69.7 ± 7.01	<0.0001	0.0046
Cardiac Function										
LVE(sec)	0.23 ± 0.03	0.25 ± 0.04	0.29 ± 0.04	<0.0001	<0.0001	0.24 ± 0.01	0.26 ± 0.03	0.3 ± 0.03	<0.0001	0.0064
LV Max (mmHg/s)	1383 ± 254	1242 ± 228	1196 ± 164	<0.0001	0.1623	1340 ± 202	1227 ± 191	1158 ± 197	0.0002	0.1722
LVC(1/s)	16.1 ± 1.5	15.4 ± 1.4	15.1 ± 1.4	0.0102	0.1734	16.0 ± 1.4	145.0 ± 1.0	14.6 ± 1.7	0.0005	0.3679
CO (L/min)	5.9 ± 1.1	5.7 ± 0.98	5.0 ± 0.5	<0.0001	<0.0001	6.2 ± 1.4	5.5 ± 1.0	5.1 ± 1.1	<0.0001	0.0039
CI (L/min/m^2^)	3.4 ± 0.6	3.3 ± 0.62	2.9 ± 0.4	<0.0001	<0.0001	3.3 ± 0.5	2.9 ± 0.4	2.7 ± 0.4	<0.0001	0.0026
SV (mL)	67.4 ± 7.3	66.8 ± 9.5	66.7 ± 7.6	0.0392	0.4108	76.0 ± 15.8	72.3 ± 10.7	72.2 ± 13.3	0.0081	0.9800
SVI (mL/m^2^)	39.1 ± 4.6	38.7 ± 5.1	38.6 ± 3.5	0.0948	0.6461	40.4 ± 5.3	38.3 ± 3.4	38.5 ± 4.7	0.0117	0.9075
**Vascular Function**										
SVC (mL/mmHg)	1.1 ± 0.2	1.2 ± 0.18	1.2 ± 0.19	0.0039	0.5188	1.3 ± 0.3	1.3 ± 0.3	1.4 ± 0.3	0.0376	0.2547
SVR(dynes/sec/cm^5^)	1325 ± 255	1344 ± 238	1485 ± 227	0.0001	<0.0001	1324 ± 341	1458 ± 305	1533 ± 379	<0.0001	0.0801
BAC (mL/mmHg)	0.07 ± 0.02	0.07 ± 0.02	0.06 ± 0.02	0.7248	0.0316	0.08 ± 0.02	0.09 ± 0.03	0.09 ± 0.02	0.1408	0.2811
BAD (%/mmHg)	5.4 ± 1.1	5.95 ± 1.10	5.5 ± 1.0	0.3507	0.0508	5.7 ± 1.1	6.6 ± 1.4	6.3 ± 1.2	0.1202	0.5035
BAR (dynes/sec/cm^5^)	179 ± 83	177 ± 78	213 ± 86	0.0066	<0.0001	152 ± 70	146 ± 68	159 ± 76	0.4939	0.1565
**SpO_2_ (%)**	83.8 ± 5.1	90.3 ± 3.4	97.7 ± 1.4	<0.0001	<0.0001	89.9 ± 3.5	92.1 ± 1.7	96.8 ± 2.1	0.0010	0.0010

**Table 5 ijerph-18-09472-t005:** Adaptation of cardiovascular function hemodynamics in high-altitude mountain in winter (N = 22).

Parameters	Tataka Baseline	Tataka after (2 h Later)	*p*_1_-Value	Lulin Villa Baseline	Lulin after (1 h Later)	*p*_2_-Value
**BP Components**						
SBP (mmHg)	136.1 ± 19.3	125.0 ± 14.4	<0.0001	141.1 ± 19.3	133.4 ± 20.8	0.0648
DBP (mmHg)	82.6 ± 10.9	80.4 ± 7.17	0.0472	85.8 ± 9.9	82.2 ± 10.7	0.2573
cSBP (mmHg)	142.9 ± 18.2	132.5 ± 16.4	0.0002	148.2 ± 18.4	141.5 ± 21.5	0.1290
cDBP(mmHg)	76.8 ± 11.0	76.4 ± 9.8	0.8511	79.8 ± 10.4	75.8 ± 10.1	0.1387
MAP (mmHg)	97.7 ± 12.4	94.4 ± 10.2	0.1193	101.3 ± 12.2	96.1 ± 12.8	0.0465
PP (mmHg)	66.1 ± 11.3	56.1 ± 12.2	0.0001	68.4 ± 1.0	65.7 ± 14.2	0.6065
HR (beats/min)	83.8 ± 10.7	82.9 ± 14.2	0.6197	88.4 ± 11.8	87.5 ± 13.0	0.2418
**Cardiac Function**						
LVE(sec)	0.24 ± 0.02	0.25 ± 0.04	0.0281	0.22 ± 0.03	0.23 ± 0.03	0.0532
LV Max (mmHg/s)	1419 ± 263	1242 ± 229	<0.0001	1463 ± 240	1383 ± 254	0.2170
LVC(1/s)	16.4 ± 1.9	15.4 ± 1.4	0.0021	16.3 ± 1.6	16.1 ± 1.5	0.4449
CO (L/min)	5.9 ± 1.0	5.7 ± 1.0	0.0206	6.0 ± 1.1	5.9 ± 1.1	0.2344
CI (L/min/m^2^)	3.5 ± 0.6	3.3 ± 0.6	0.0170	3.6 ± 0.7	3.4 ± 0.6	0.2208
SV (mL)	69.4 ± 9.8	66.8 ± 9.5	0.0100	67.5 ± 6.8	67.4 ± 7.3	0.5578
SVI (mL/m^2^)	40.3 ± 5.1	38.7 ± 5.1	0.0064	39.7 ± 4.7	39.1 ± 4.6	0.5501
**Vascular Function**						
SVC (mL/mmHg)	1.1 ± 0.18	1.2 ± 0.2	0.0006	1.0 ± 0.1	1.1 ± 0.2	0.3606
SVR(dynes/sec/cm^5^)	1351 ± 245	1345 ± 238	0.8452	1376 ± 263	1326 ± 255	0.8754
BAC (mL/mmHg)	0.06 ± 0.02	0.07 ± 0.02	0.0610	0.07 ± 0.02	0.07 ± 0.02	0.7425
BAD (%/mmHg)	5.3 ± 1.0	6.0 ± 1.1	0.0104	5.1 ± 0.8	5.4 ± 1.11	0.3198
BAR (dynes/sec/cm^5^)	207 ± 104	177 ± 78	0.0514	175 ± 80	179 ± 83	0.3215
**SpO_2_ (%)**	88.17 ± 5.46	90.3 ± 3.4	0.1282	81.2 ± 8.0	83.8 ± 5.1	0.6052

**Table 6 ijerph-18-09472-t006:** Generalized linear mixed-effect models for determinants of seasonal variations on blood pressure and cardiovascular function.

	High alt. vs. Low alt.		High alt. vs. Low alt.	
Variables	in Winter		in Summer	
	Est. ± S.E.	*p*	Est. ± S.E.	*p*
**BP Components**				
SBP (mmHg)	9.16 ± 3.40	0.0088	3.93 ± 6.14	0.5242
DBP (mmHg)	3.28 ± 2.23	0.1093	−0.86 ±3.65	0.8148
cSBP (mmHg)	7.61 ± 3.51	0.0334	4.47 ± 6.33	0.4820
cDBP(mmHg)	2.71 ± 2.23	0.2288	−0.23 ± 4.03	0.9551
MAP (mmHg)	4.23 ± 2.36	0.0780	2.92± 4.26	0.4961
PP (mmHg)	5.02 ± 2.89	0.0865	4.30 ± 5.21	0.4118
HR (beats/min)	16.48 ± 2.35	<0.0001	9.51 ± 4.24	0.0282
**Cardiac Function**				
LVE(sec)	−0.06 ± 0.01	<0.0001	−0.04 ± 0.02	0.0511
LV Max (mmHg/s)	207.42 ± 60.51	0.0010	178.14 ±105.53 0.03	0.0959
LVC(1/s)	1.86 ± 0.55	0.0012	2.07 ± 0.96	0.0346
CO (L/min)	1.42 ± 0.26	<0.0001	1.30 ± 0.46	0.0060
CI (L/min/m^2^)	0.83 ± 0.14	<0.0001	0.64 ± 0.24	0.0101
SV (mL)	2.98 ± 2.03	0.1468	7.0 ± 3.6	0.0531
SVI (mL/m^2^)	1.89 ± 1.10	0.0912	3.28 ± 1.93	0.0934
**Vascular Function**				
SVC (mL/mmHg)	−0.07 ± 0.06	0.2669	0.01 ± 0.11	0.9150
SVR(dynes/sec/cm^5^)	−350.98 ± 70.21	<0.0001	−185.91 ± 122.73	0.1344
BAC (mL/mmHg)	−0.004 ± 0.005	0.4228	−0.004 ± 0.008	0.6286
BAD (%/mmHg)	−0.39 ± 0.40	0.3259	−0.34 ± 0.69	0.6227
BAR (dynes/sec/cm^5^)	−43.49 ± 12.33	0.0008	−12.38 ± 21.51	0.5667
**SpO_2_ (%)**	−8.25 ± 1.47	<0.0001	−7.98 ± 2.84	0.0070

## Data Availability

The data presented in this study are available on request from the corresponding author.

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
