# Peer review of "Seasonal Effects of High-Altitude Forest Travel on Cardiovascular Function: An Overlooked Cardiovascular Risk of Forest Activity"

_ijerph, 2021, doi:10.3390/ijerph18189472_

Round 1
Reviewer 1 Report
I wonder if there is any literature which may be cited on very fit individuals under physical stress in mountain environments (cyclists, mountain climbers). If so this might be cited and some note made of the differences from these subject groups.
I might note that these groups were in some degree in poor physical health:
"Hypertensive adults taking med- 213
ication accounted for 18.2% of the participants, and 6.3% were diagnosed with hypercho- 214
lesterolemia. "
Additionally an ethical concern here, albeit I am sure that the researchers were conscious of the health conditions of the subjects. Some short discussion on appropriateness of the subjects might be noted and their recruitment.
The setting was that of a forested mountain area. Is the fact that it is forested of any significance? It might be noted that symbolically forests visits impart of strong sense of health giving (ie 'forest bathing) which might be a cultural factor present which clouds the risk perceptions of forest walking at high altitude.
Author Response
Response to the Review #1 Comments
I would like to thank for your valuable comments. The manuscript has been revised accordingly. I hope that the revised manuscript would meet with your approval for publication in IJERPH. Authors’ replies to the specific comments are stated below.
Reviewer #1 Comments and Suggestions for Authors
- I wonder if there is any literature which may be cited on very fit individuals under physical stress in mountain environments (cyclists, mountain climbers). If so this might be cited and some note made of the differences from these subject groups. Reply: Thanks for your inspiring comments. We found the following literature which conducted their investigations under similar conditions as our study (rapid ascension from low to high altitude).
Case 1
Twenty-seven healthy normotensive subjects (8 children, 9 adults, and 10 elderly subjects) of a wide age range (6-83 years) were recruited and had a normal body mass index. None of the subjects were smokers, taking drugs, or affected by hypertension or diseases that could influence cardiovascular and autonomic nervous function.
All measurements were performed at the subjects' home town, Turin, Italy, about 200 m above sea level (basal condition) and repeated at 2950 m (Mosso Institute, Gressoney la TrinitY, Aosta, Italy). All subjects were transported to high altitude via cable car in 30 minutes. Environmental temperature was lower at 2950 m (2-12°C) than at 200 m (27-32°C). 24-h ambulatory noninvasive BP recordings, using an oscillometric monitor. BP was measured every 15 minutes during daytime and every 30 minutes during sleep. The results showed that the BP and HR of all subjects significant increased exposure from lower altitude (200 m) to high altitude (2950 m) rapid ascent by cable car in 30 minutes (Ref.19 of revise manuscript).
Case 2
A group of 139 healthy young males from 500 m rapid ascent by train, the HR, ejection fraction, fractional shortening, SV, and CO were significantly increased within 24 hours of arrival at 3700 m. The HR, SBP, DBP, and mean artery pressure were significantly increased on the 7th day of acclimatization at 4400 m high altitude to compare with 3700 m of baseline level in all subjects (Ref. 3 of revised manuscript).
Such study shows that regardless of whether it is a healthy subject or a patient with hypertension and hyperlipidemia in this study, rapid ascension from low to high altitude often causes the risk of cardiovascular load.
The content of case 1 and 2 has been included in the discussion of revised manuscript, please see the revised manuscript.
2. I might note that these groups were in some degree in poor physical health:
"Hypertensive adults taking medication accounted for 18.2% of the participants, and 6.3% were diagnosed with hypercholesterolemia."
Reply:
Thanks for your inspiring comments. The hypertension and hypercholesterolemia subjects are treated at National Taiwan University Hospital, and I am their attending physician. I accompanied the subjects to participate high-altitude studies in winter and summer seasons. Their safety is under control.
3. Additionally an ethical concern here, albeit I am sure that the researchers were conscious of the health conditions of the subjects. Some short discussion on appropriateness of the subjects might be noted and their recruitment.
Reply:
Thanks for your valuable comments. The hypertension and hypercholesterolemia subjects were recruited for an observational pilot study. These participants have provided their written informed consent before receiving a series of detailed examinations and questionnaires. This research has been approved by Research Ethics Committee of the National Taiwan University Hospital (201704031RIND).
4. The setting was that of a forested mountain area. Is the fact that it is forested of any significance? It might be noted that symbolically forests visits impart of strong sense of health giving (i.e. ‘forest bathing) which might be a cultural factor present which clouds the risk perceptions of forest walking at high altitude.
Reply:
Thanks for your comments. We agree that symbolically forests visits impart of strong sense of health giving (i.e. forest bathing), which clouds the risk perceptions of forest walking at high altitude. It is generally believed that forest bathing has positive health benefits, which also confirmed from many scientific research reports. Most people will take forest bathing activities in the low-altitude mountain forests near their homes. However, high-altitude activity involves traveling from a low altitude flat land to a medium-high altitude area of 2500-3500 meters became more and more popular in recent decades. Rapid ascension from low to high altitude often causes acute mountain sickness. High cardiovascular stress associated with altitude-environment changes is an important emerging public health issue. People often ignore the risk of cardiovascular load caused by hypoxia and low temperature in high-altitude mountainous areas. We believe that our study findings can provide readers for understanding the risk perceptions of forest walking at high altitude hypoxia and low temperature environments.

Reviewer 2 Report
In this study authors evaluated Seasonal Effects of High-Altitude Forest Travel on Cardiovascular Function.
In Introduction section they presented and described the changes in cardiovascular system with change in altitude and highlighted importance of cardiovascular response particularly at high altitude including mountain sickness. The aim is well formulated with proper analysis for both seasons including winter and summer.
In material section the groups that were gathered for this investigations were somehow of low number of participants, but it was noticed in limitations section, and those for winter study had 2 days and those from summer group one day program. The tested parameters and study design aside this is adequate.
Cardiac and Vascular functional assessment are adequate. Also statistical analysis is fully explained.
In results section the tested groups seemed to by different in various parameters...diabetes, smoking habits...hypertension...etc. Maybe it would be of interest to add in Table 1 the number of comorbidities as well.
The tables were well described in results text.
In discussion authors pointed to significant increase in cardiovascular workload from low to high altitude forest in acute high altitude travelers, with additional parameters of BP and cardiac workload that were in certain areas significantly higher. Such claims authors adequately elaborated and compared with previous findings. Additional advantage of this study is also stated in terms of positive effects of study findings on future public health actions and physiological need of human body in high altitude hypoxia environment.
Study limitations are adequate.
Author Response
I would like to thank for your valuable comments. High-altitude activity involves traveling from a low altitude flat land to a medium-high altitude area of 2500-3500 meters became more and more popular in recent decades. Rapid ascension from low to high altitude often causes acute mountain sickness. High cardiovascular stress associated with altitude-environment changes is an important emerging public health issue. Cardiovascular physiological responses involving hypoxemia in cold temperature environments at high altitude have yet to be adequately investigated. We believe that study findings can provide readers for understanding the risk perceptions of forest walking at high altitude hypoxia and low temperature environments.

Reviewer 3 Report
Dear Authors,
It is a very interesting article, it provides relevant knowledge about cardiovascular and homeostatic functions in general when the body is exposed to different heights, climatic seasons and temperatures. This knowledge is transferable towards the health of the people who make excursions in the mountains. It is a very complete work in terms of its content and structure.
Consider just a few things:
The abstract should describe the study participants.
The abstract must have a maximum of 200 words. Correct it.
Do the authors believe that there is obsolescence in the data? (line 104).
In my opinion, the paragraph between lines 145 and 158 could be deleted.
Thanks.
Author Response
Response to the Review #3 Comments
I would like to thank you for the valuable comments and suggestions. The manuscript has been revised with due attention to the reviewers’ comments. Authors’ replies to the comments are stated below.
Reviewer #3 Comments and Suggestions for Authors
It is a very interesting article, it provides relevant knowledge about cardiovascular and homeostatic functions in general when the body is exposed to different heights, climatic seasons and temperatures. This knowledge is transferable towards the health of the people who make excursions in the mountains. It is a very complete work in terms of its content and structure.
Consider just a few things:
1. The abstract should describe the study participants.
Reply: I would like to thank for your valuable comments. We added additional details to describe the study participants in the abstract of revised manuscript.
2. The abstract must have a maximum of 200 words. Correct it.
Reply: I would like to thank for your valuable comments. We revised the abstract accordingly.
3. Do the authors believe that there is obsolescence in the data? (line 104).
Reply: I would like to thank for your valuable comments. We attempted to correct the potential obsolescence in the data through statistical analysis with mixed effects model. We are also cautious not to over-interpret the results.
4. In my opinion, the paragraph between lines 145 and 158 could be deleted.
Reply: I would like to thank for your valuable comments. We deleted the paragraph between lines 145 and 158, please see the revised manuscript.
